# Transcriptomic Profiling for the Autophagy Pathway in Colorectal Cancer

**DOI:** 10.3390/ijms21197101

**Published:** 2020-09-26

**Authors:** Justyna Gil, Paweł Karpiński, Maria M. Sąsiadek

**Affiliations:** 1Department of Genetics, Wroclaw Medical University, Marcinkowskiego 1, 50-368 Wroclaw, Poland; pawel.karpinski@umed.wroc.pl (P.K.); maria.sasiadek@umed.wroc.pl (M.M.S.); 2Laboratory of Genomics & Bioinformatics, Institute of Immunology and Experimental Therapy, Polish Academy of Sciences, 53-114 Wroclaw, Poland

**Keywords:** autophagy, gene expression, colorectal cancer

## Abstract

The role of autophagy in colorectal cancer (CRC) pathogenesis appears to be crucial. Autophagy acts both as a tumor suppressor, by removing redundant cellular material, and a tumor-promoting factor, by providing access to components necessary for growth, metabolism, and proliferation. To date, little is known about the expression of genes that play a basal role in the autophagy in CRC. In this study, we aimed to compare the expression levels of 46 genes involved in the autophagy pathway between tumor-adjacent and tumor tissue, employing large RNA sequencing (RNA-seq) and microarray datasets. Additionally, we verified our results using data on 38 CRC cell lines. Gene set enrichment analysis revealed a significant deregulation of autophagy-related gene sets in CRC. The unsupervised clustering of tumors using the mRNA levels of autophagy-related genes revealed the existence of two major clusters: microsatellite instability (MSI)-enriched and -depleted. In cluster 1 (MSI-depleted), *ATG9B* and *LAMP1* genes were the most prominently expressed, whereas cluster 2 (MSI-enriched) was characterized by *DRAM1* upregulation. CRC cell lines were also clustered according to MSI-enriched/-depleted subgroups. The moderate deregulation of autophagy-related genes in cancer tissue, as compared to adjacent tissue, suggests a prominent field cancerization or early disruption of autophagy. Genes differentiating these clusters are promising candidates for CRC targeting therapy worthy of further investigation.

## 1. Introduction

Colorectal cancer (CRC) is among the most frequent cancers (3rd) and is one of the major causes of cancer-related deaths (4th) worldwide [1,2]. In analyses of the time period from 2000 to 2010, it has been shown that age-standardized cancer mortality rates (ASMRs) decreased in most countries, especially in the USA [3,4]. However, in many countries, despite great progress in both cancer screening (e.g., capsule endoscopy and/or fecal occult blood testing) and early cancer diagnosis, racial and socioeconomic disparity remains, leading to late cancer diagnosis and hence to a low level of 5-year survival [1]. Interestingly, the past two decades have shown that CRC incidence among younger populations (less than 50 years of age) has increased, especially in well-developed, highly industrialized countries [5,6]. The majority of CRC cases (more than 70%) are sporadic; therefore, the combination of an individual’s genetic makeup, exposure to environmental factors, and personal habits modulate the risk of colorectal cancer development [7].

The classification of cancer subtypes based on gene expression is widely accepted as an important component of disease taxonomy [8,9,10,11]. Integrated genome-wide studies have revealed that the single-tissue cancer type may be divided into several molecular subtypes. Definitions of cancer subtypes are undoubtedly required for the implementation of individual effective therapy [8,9,10,11]. Recently, an international consortium consisting of expert groups divided CRCs into four consensus molecular subtypes (CMS1–CMS4), each of which presents with distinct biological features and specific gene expression patterns. This taxonomy implements self-contained prognostic information that transcends tumor stages and tissue of origin. Further studies may provide new pathway-based insights with directions for treatment optimization [8,12].

In this aspect, macroautophagy (henceforth referred to as autophagy) appears to be a promising process for seeking new biomarkers and thus may serve to enhance our knowledge of potential therapeutic molecular targets in CRC treatment. Autophagy is a highly evolutionary conserved housekeeping process that relies on the degradation of cellular components (e.g., organelles, protein aggregates) in autophagosomes. Altered activity in this pathway (compared to normal conditions) is associated with numerous human diseases, including cancer [13,14]. Autophagy has a multifaceted nature, and its role in cancerogenesis from the initiation of tumor formation to metastasis may have opposite consequences in a context-dependent manner [15,16]. On the one hand, enhanced autophagy allows the elimination of cells susceptible to becoming malignant (by the elimination of potentially tumor prone factors) because of its “repair” function and thus inhibits cancer development and acts as a tumor suppressor. This view is supported by the observation of monoallelic deletion of the *BECN1* gene that has been found to be common in various human cancers such as breast, prostate, or ovarian [17,18]. Consequently, insufficient autophagy may contribute to the accumulation of genomic defects, leading to cancer development [14]. On the other hand, in developed tumor cells that are often exposed to unfavorable conditions such as hypoxia, starvation, or stress induced by treatment (chemo-/radiotherapy), enhanced autophagy may help these cells to survive (as supplier of recycled molecules crucial for survival); hence, it acts as a tumor promoter [19]. Therefore, elucidating the autophagy process in cancer development remains a great challenge also in the context of anticancer treatment. In the therapy aspect, there are two suggested approaches: in the first one, autophagy is targeted by well-known inhibitors such as hydroxychloroquine (to stop a supportive influence of autophagy by promoting apoptosis); in the second one, autophagy is induced (to enhance the antitumor therapy by the induction of autophagy-dependent cell death, a cytotoxic role) by, for instance everolimus or perifosine [20,21,22]. Hitherto, many clinical trials have been carried out with promising results. The above-mentioned agents were usually well tolerated along with standard treatment, but further studies are needed to clarify the role of autophagy and its modulation in cancer treatment.

As a result of the complex and unclear role of autophagy in CRC, we performed an analysis of transcriptomic data of chosen autophagy genes. Our goal was to define the contribution of autophagy (in terms of gene expression) in colorectal cancer samples in comparison to adjacent tissue and to investigate a potential specific pattern to identify a new molecular CRC subtype that might be helpful in the molecular classification and/or in the development of future treatment approaches.

## 2. Results

### 2.1. Ensemble of Gene Set Enrichment Analysis (EGSEA)

To evaluate the involvement of autophagy in the process of CRC carcinogenesis, we preformed gene set enrichment analysis in a CRC RNA sequencing (RNA-seq) dataset by comparing 2055 gene sets in cancerous vs. normal tissue using the Ensemble of Gene Set Enrichment Analysis (EGSEA) approach. Results are provided in Appendix A. Five out of six autophagy-related gene sets were significantly deregulated (up or downregulated) in colorectal cancer when compared to normal tumor-adjacent tissues. However, autophagy-related gene sets were relatively low ranked when compared to other gene sets. For example, “Late endosomal microautophagy” reached the highest ranking among autophagy gene sets and was ranked 1305/2055. The other significantly deregulated autophagy-related gene sets were ranked as follows 1413, 1493, 1815, and 1854 (see Appendix A). 

### 2.2. Unsupervised Clustering of Colorectal Tumor Samples and Cell Lines

Next, we posed a question of whether there are subgroups of CRC differing in terms of the expression of selected autophagy genes (genes are listed in Appendix A). In all three studied datasets (CRC RNA-seq, CRC array, and CRC cell lines), samples were separated into two clusters (optimal k = 2) based on the expression of autophagy genes (defined as clusters 1 and 2). Table 1 and Table 2 provide characteristics of resultant CRC clusters. In the RNA-seq tumor dataset, 46% of samples belonged to cluster 1 and 54% belonged to cluster 2. In the Affymetrix tumor dataset, 41% of samples belonged to cluster 1 and 59% belonged to cluster 2. In the CRC cell line dataset, 31% of samples belonged to cluster 1, and 69% belonged to cluster 2 (data not shown). In general, we found cluster 1 to be depleted in (MSI) samples, whereas, in comparison, cluster 2 was significantly enriched in MSI samples.

### 2.3. Molecular Characteristics of Autophagy Clusters (RNA-seq Data)

Table 1 provides detailed molecular characteristics of resultant CRC clusters in RNA-seq data. Cluster 1 was associated with a higher aneuploidy score, homologous recombination defects, and a higher frequency of *APC* and *TP53* mutations (all *p*-values < 0.001), whereas cluster 2 was associated with a higher frequency of microsatellite instability, proximal tumor location, *BRAF* mutations, single nucleotide variant neoantigens, indel neoantigens, and non-synonymous mutations (all *p*-values < 0.001). We observed differences in the distribution of CMS subtypes and stages across two autophagy clusters. Higher frequencies of CMS1 (25 vs. 9%) and CMS3 (24 vs. 10%) were noted in cluster 2, whereas cluster 1 was significantly enriched in CMS2 (52 vs. 14%). A slightly greater number of stage 1 tumors were noted in cluster 2 than in cluster 1 (12 vs. 21%, respectively). Stage 4 tumors were observed more frequently in cluster 1 than in cluster 2 (20 vs. 12%, respectively). No significant differences between clusters were detected for overall survival (*p*-value = 0.55, data not shown), age, gender, or mutation frequency of *KRAS*, *NRAS*, and *PIK3CA*.

### 2.4. Molecular Characteristics of Autophagy Clusters (Affymetrix Data)

Table 2 provides the molecular characteristics of resultant CRC clusters in array data. Cluster 2 was associated with a higher frequency of microsatellite instability and a higher age of morbidity. We observed differences in the distribution of CMS subtypes and stages across two autophagy clusters. Higher frequencies of CMS1 (25 vs. 7%) and CMS4 (40 vs. 25%) were noted in cluster 2, whereas cluster 1 was significantly enriched in CMS2 (50 vs. 16%). No significant differences were noted between autophagy clusters in terms of overall survival (*p*-value = 0.35, data not shown) or stage and gender distribution.

### 2.5. Selection of the Autophagy Genes that Best Define Autophagy Clusters

With the use of sparse partial last squares discriminant analysis PLS (sPLS-DA), we attempted to define which of the autophagy genes best differentiate CRC samples between two autophagy clusters. Results from three datasets are presented in Figure 1 and Figure 2. Sixteen genes were selected in the array dataset (Figure 1A), and nine genes were selected based on RNA-seq analysis (Figure 1B). Four genes (*ATG16L2*, *ATG9B*, *DRAM1*, and *LAMP1*) were found to be common to both datasets. The most highly ranked genes in terms of separation of clusters were *ATG9B*, and *DRAM1* (Figure 1C,D)*. ATG9B* and *LAMP1* were characterized by the highest expression level in cluster 1, while *DRAM1* was characterized by the highest expression level in cluster 2. In the CRC cell lines dataset, 16 genes were selected by sPLS-DA (Figure 2A). The most highly ranked genes in terms of separation of clusters were *UVRAG* and *ATG4C* (Figure 2B).

### 2.6. Differential Expression (DE) of Autophagy Genes

To illustrate differences in the expression of autophagy genes in resultant autophagy clusters, we performed differential expression analysis by comparing the transcriptome of each cluster with normal-adjacent tissues. This has been done in CRC RNA-seq and CRC array datasets. The results of DE analysis are presented in Figure 3, and lists of genes with important parameters are provided in Appendix A. In general, the detected effects in two CRC clusters were relatively small, not exceeding absolute log fold change (logFC) = 1, with the exception of *ATG4A, ATG4D, ATG9B, DRAM1, GABARAPL1,* and *ULK4*. For these genes, absolute logFC values were above 1; however, it was not consistent across datasets and/or clusters except for *DRAM1*, for which logFC > 1 was noted in cluster 2 in both datasets. Clusters 1 and 2 were similar in terms of the direction of gene expression. Fifteen genes in both clusters and both datasets were downregulated: *ATG4D*, *ATG4A*, *CALCOCO2*, *GABARAPL1*, *ULK3*, *WIPI1*, *MAP1LC3B2*, *SH3GLB1*, *GABARAP*, *BECN1*, *GABARAPL2*, *MAP1LC3B*, *ATG5*, *DRAM2*, and *ATG13.* Six genes exhibited upregulation in both clusters and both datasets: *LAMP1*, *SQSTM1*, *WIPI2*, *VCP*, *DRAM1*, and *ATG9B.* As for cluster-specific deregulation, *ATG4C* displayed cluster-1-specific downregulation, whereas no gene was specifically downregulated in cluster 2. Furthermore, none of the genes showed cluster-specific upregulation.

## 3. Discussion

Currently, the molecular characteristics of highly heterogeneous CRC constitute a medical standard for the prognosis, prediction, and selection of the most effective treatment approach [23]. As the survival ratio of CRC patients remains unsatisfactory (especially in groups of patients at advanced stages), new biomarkers may enable the implementation of novel anticancer strategies that target pathways other than the classic apoptosis–death. Recently, autophagy has been extensively studied in different types of cancer in order to obtain a better understanding of its involvement in both the molecular process of carcinogenesis and treatment resistance. It is postulated that the modulation (induction and inhibition) of autophagy may be essential for the improvement of anticancer adjuvant therapy [24,25].

EGSEA analysis preformed in this study revealed the significant deregulation of autophagy-related pathways in CRC. In general, autophagy gene sets were lowly ranked in EGSEA analysis, which was possibly due to the moderate gene expression changes that we observed in DE analysis. This explains why autophagy has been often overlooked in gene set enrichment results provided for CRC or CRC clusters [8]. However, moderate gene expression changes may still represent a biologically meaningful factor [26]. Our study, based on the expression pattern of autophagy-related genes in colon tumors vs. tumor-adjacent normal tissue, using two large RNA-seq and microarray datasets, revealed the existence of two separate clusters. In terms of clinical and molecular characteristics, the two clusters differ significantly, suggesting the existence of two groups of CRCs: cluster 1, presenting with prominent chromosomal instability, distal tumor localization, and high *TP53* and *APC* mutation rates, and cluster 2, with a high mutation rate, proximal tumor localization, and microsatellite instability. This result has been confirmed in the CRC cell lines dataset with respect to MSI status.

Interestingly, in the aspect of widely accepted CMS clustering, our study showed that cluster 1 covers the majority of CMS2 and CMS4, while cluster 2 covers the majority of CMS1 and CMS3. These results are in agreement with the biological nature of CMSs and may be used as an additional biomarker for CRC subtyping and selecting therapeutic approaches. As revealed by differential expression analysis, the two clusters were similar in terms of the direction of expression changes of significantly deregulated genes. Importantly, most expression changes were of little magnitude. This result may indicate that changes in the expression of autophagy genes in adjacent normal tissue are relatively numerous, suggesting a prominent field defect or the early disruption of autophagy in CRC. Similarly, a recent study by Aran et al. has demonstrated major gene expression changes in tissues adjacent to tumors [27]. Therefore, we suggest that future studies on changes in gene expression in autophagy-related genes should employ normal colon tissue dissected from individuals free of cancer.

The machine learning approach (sPLS-DA) indicated that the two clusters in tumor samples differ with respect to the expression levels of several autophagy-associated genes: cluster 1, with the prominent expression of *ATG9B* along with *LAMP1*, and cluster 2, with high levels of *DRAM1*. Intriguingly, the genes differentiating these two clusters are closely associated with lysosomal degradation.

LAMP1 (lysosomal associated membrane protein 1) is a highly glycosylated protein associated with the regulation of lysosome mobility and its fusion with the autophagosome membrane [28]. LAMP1 is expressed on (apart from the lysosomal membrane) the cell surface of many different human cells [29]. The overexpression of LAMP1 has been linked to high-grade breast tumors capable of metastasis; hence, its high expression appears to be associated with malignant attributes [30]. Recent studies have shown LAMP1 overexpression of both mRNA and protein levels in high-grade glioblastoma multiforme (GBM) [31]. Moreover, high expression levels of LAMP1 in the plasma membrane were observed, inter alia, in colorectal neoplasm tissues, advanced prostate cancer (PCa), and castration-resistant prostate cancer (CRPC) [29,32]. Recently, Takeda et al. demonstrated that the number of autophagy genes including *LAMP1* is effectively downregulated by mefloquine hydrochloride (an antimalarial drug) in colorectal cell lines and PDX models, and it has a strong negative effect on cancer cells [33].

Another highly expressed gene in cluster 1 was *ATG9B*, encoding the multi-spanning transmembrane protein that plays a key role in the biogenesis of autophagosome membranes [34]. Kang et al. have shown a high rate of *ATG9B* mutations in human high-MSI gastric and colorectal cancers. It has been postulated that the deregulation of ATG9B may contribute to promotion of the development of stomach and colorectal cancers [34].

In the second cluster, *DRAM1* (DNA damage regulated autophagy modulator 1) is the most expressed gene. DRAM1 is a lysosomal membrane protein that participates in the process of autophagosome–lysosome fusion [35], and thus, its expression promotes autophagosome formation. It has been shown that in response to adverse genotoxic stimuli, *TP53*, a major tumor-suppressing gene responsible for apoptotic cell death, induces autophagy via the transcriptional activation of DRAM1 [36]. Crighton et al. have demonstrated that DRAM1 is part of the network of cell death pathways dependent on TP53, which simultaneously activates DRAM1 and other unknown proapoptotic gene/genes, which in turn contribute together to the cell-death response. It is also possible that the autophagic and apoptotic potential of DRAM1 is separable and acts in parallel [36].

Since *DRAM1* is a target gene for TP53, it is critical in TP53-induced apoptosis. DRAM1 influences apoptosis via lysosomes thorough the proapoptotic BAX gene. DRAM1 improves the translocation of BAX to lysosomes to initiate lysosome-dependent apoptotic death [37]. Therefore, it is suggested that DRAM1 may connect the autophagy and apoptosis pathways. The induction of autophagy by TP53 via DRAM1 contributes to apoptotic cell death, which appears to be crucial for the development of novel standards of anticancer treatment.

Recently, many research groups have been searching for new candidate genes related to autophagy (as a prognostic and/or prediction biomarkers) in CRC using available datasets [38,39]. We emphasize that our study differs especially with respect to the level of gene selection. In databases, in the section of autophagy-related genes, different genes that participate in autophagy solely are included along with genes from other important pathways that overlap with the autophagy pathway under certain conditions. We claim that naming *VEGFA*, *NRG1,* or *CDKN2A* as autophagy-related genes is at least misleading, because their major role does not involve the autophagy pathway [38]. Hence, to avoid ambiguous results, the genes chosen for this study include those that are highly engaged in the autophagy pathway.

In the current study, we were unable to demonstrate the intersection between tumor and cell lines in relation to deregulated autophagy genes. This may reflect differences between tumor samples and cell lines. In the latter, an intact immune system and stroma are lacking, and specific lineage subtypes are overrepresented. Indeed, Koustas et al. and Folkerts et al. recently discussed the close interplay between autophagy, tumor environment, and immune response [40,41], thus demonstrating the limitations of cell lines as a model of autophagy in CRC. Nevertheless, our study clearly showed that the expression of autophagy-related genes in CRC displays two distinctive patterns for two different subgroups, i.e., MSI-enriched and MSI-depleted. Given that MSI-enriched CRCs display strong infiltration of the tumor microenvironment with immune cells, the differences observed in this study may be linked to autophagic processes in the tumor microenvironment rather than to the epithelial fraction of the tumors. This may be supported by the notion that autophagy is a key factor in various immune responses against tumors, including antigen presentation and T-cell activation [42].

## 4. Study Limitations

There are potential limitations to this study that should be considered. A relatively small control group is available in databases (normal tissues or adjacent normal tissues). The most convenient setting for statistical analysis is when the sample size is comparable in terms of gender and age. Hence, more population/epidemiological data on RNA profiles (transcriptome) are required. To our best knowledge, there is a lack of access to proteomic data concerning autophagy in colorectal cancer. This situation impeded a comprehensive analysis of protein levels that undoubtedly would raise the power of this analysis. Further studies on protein levels are needed as well as accessible protein databases. 

The goal of the current study was not to determine if autophagy plays a suppressing or promoting function in CRC because of its context-dependent manner. This study is descriptive, and unfortunately, we are a long way from this definition being made.

## 5. Materials and Methods

### 5.1. Data Acquisition

RNA-seq data (raw counts) for colon and rectal adenocarcinoma (CRC, 620 samples) were obtained along with tumor-adjacent tissue (53 samples) from compiled RNA-Sequencing TCGA data published by Rahman et al. under accession number GSE62944 in Gene Expression Omnibus (GEO) [43].

Raw CEL (.cel) files (Affymetrix U133 Plus 2.0 microarray expression)(Affymetrix, Santa Clara, CA, USA) were collected from 15 studies that included 1597 tumor and 125 normal colon samples (Gene Expression Omnibus accession numbers: GSE69657, GSE8671, GSE9254, GSE13067, GSE13294, GSE14333, GSE17536, GSE17537, GSE18105, GSE19860, GSE28702, GSE33113, GSE35896, GSE37364, and GSE39582 [44,45,46,47,48,49,50,51,52,53,54,55,56]).

Non-synonymous mutation data (TCGA dataset only) were obtained from Genomic Data Commons. Clinical and molecular data, including overall survival and microsatellite instability status, were obtained from accompanying phenotype files specific to each study. Other variables, including single nucleotide variants and non-silent mutation and aneuploidy scores, were obtained from Thorsson et al. [57].

Raw CRC files (.crc) of cell lines (Affymetrix U133 Plus 2.0 microarray expression) were downloaded from Array Express (E-MTAB-3610) [58].

### 5.2. Consensus Molecular Subtyping

In order to assess consensus molecular subtypes (CMS) for CRC samples, we used the nearest template prediction (NTP) algorithm implemented in the CMScaller package with default settings using normalized data [59]. Samples with false discovery rate adjusted *p*-values > 0.05 were designated “not assigned” (NA) and removed from subsequent analysis.

### 5.3. Data Pre-Processing

RNAseq data for CRC and tumor-adjacent tissues data were obtained from GSE62944 [43]. Subsequently, lowly expressed genes were removed by means of filtering. Next, data were normalized using variance-stabilizing transformation and smooth quantile normalization. Outlier samples were detected and removed automatically employing PCDIST algorithm [60].

Affymetrix raw data were normalized by Robust Multi-array Average (RMA) using the “affy” package, mapped to the NCBI Entrez Gene identifiers using a custom chip definition file (Brainarray, Version 20) [61,62]. Outlier samples were detected and removed automatically using PCDIST algorithm [60]. For both datasets, we used the ComBat algorithm implemented in the “swamp” package to correct the data for batch effects [63].

### 5.4. Selection of Genes of Interest

We carried out extensive searches of the literature and databases (for example, Human Autophagy Database: http://autophagy.lu/clustering/index.html) in order to select important genes involved in the autophagy pathway. We attempted at selecting genes with roles non-overlapping with other pathways. Our final list comprises 46 genes involved in autophagy (Appendix A).

### 5.5. Ensemble of Gene Set Enrichment Analysis

The RNA-seq dataset was used to analyze gene set enrichment. In brief, we applied the Ensemble of Gene Set Enrichment Analyses (EGSEA) method for gene set testing in the RNA-seq dataset by comparing tumor versus tumor-adjacent tissue [64]. We utilized the analysis of four prominent GSE algorithms: camera, fry, ora, and z-score to calculate the collective significance scores for each gene set (we used 2055 Human Reactome definitions from January 2020) [65]. Gene sets were ordered by median ranking (median across the ranks assigned by four selected methods). 

### 5.6. Unsupervised Clustering of Autophagy Genes

We used the (COMMUNAL—Combined Mapping of Multiple clUsteriNg ALgorithms) approach to provide robust clusters based on the expression of selected autophagy genes [66]. The optimal number of clusters was deduced based on integrative analysis of three clustering algorithms and nine cluster validity metrics across increasing autophagy gene subsets: 12, 24, and 36.

### 5.7. Selection of the Autophagy Genes that Best Define Autophagy Clusters

We applied sparse partial least squares discriminant analysis (sPLS-DA) to obtain the most relevant autophagy genes differentiating between autophagy clusters. Each dataset (RNAseq and Affymetrix) was analyzed separately with the use of a sPLS-DA classifier that had been trained and evaluated using 10-fold cross-validation repeated 100 times. The genes selected most frequently by the sPLS-DA classifier (frequency ≥ 0.9) were selected for the drawing of heat maps in the Complex Heatmap package.

### 5.8. Differential Expression (DE) Analysis

For array-based data, differential expression between study groups was assessed by applying a moderated t-test implemented in the “limma” package. For the RNAseq data, differential expression between study groups was assessed by applying the voom method.

*p*-values were adjusted using overall comparisons employing the Benjamini–Hochberg (BH) method. Gene expression with a *p*-value ≤ 0.05 after the BH correction was considered significantly deregulated. All available genes were included in this analysis.

### 5.9. Comparison of Expression of Selected Autophagy-Related Genes in CRC Cell Lines

Differences in gene expression between cell line clusters were assessed by applying a t-test (for normal distribution) or non-parametric Kruskal–Wallis test (for non-normal distribution). All reported *p*-values were corrected for multiple testing using the BH method.

### 5.10. Molecular Characteristics of Clusters

Differences between clusters in terms of categorical variables were assessed using a chi-squared test. Continuous variables were assessed for distribution (normal, non-normal) using the Shapiro–Wilk test. Subsequently, we assessed differences between subtypes using a t-test (for normal distribution) or non-parametric Kruskal–Wallis test (for non-normal distribution). All reported *p*-values were corrected for multiple testing using the BH method. The workflow of the current study is summarized in Appendix A.

## 6. Conclusions

We found a moderate deregulation of autophagy-related genes in CRC compared to adjacent tissue. This may indicate that changes in the expression of autophagy genes in adjacent tissue are relatively high, thus suggesting a prominent premalignant field defect or early disruption of autophagy. This implies that future studies on changes in the expression of autophagy-related genes should employ normal colon tissue dissected from individuals free of cancer. Moreover, our study identified two autophagy sets of genes (modules) in CRC: MSI-enriched and MSI-depleted. Genes that differ between clusters are connected to the lysosomal degradation pathway and may constitute candidates for CRC targeting therapy.

## Figures and Tables

**Figure 1 ijms-21-07101-f001:**
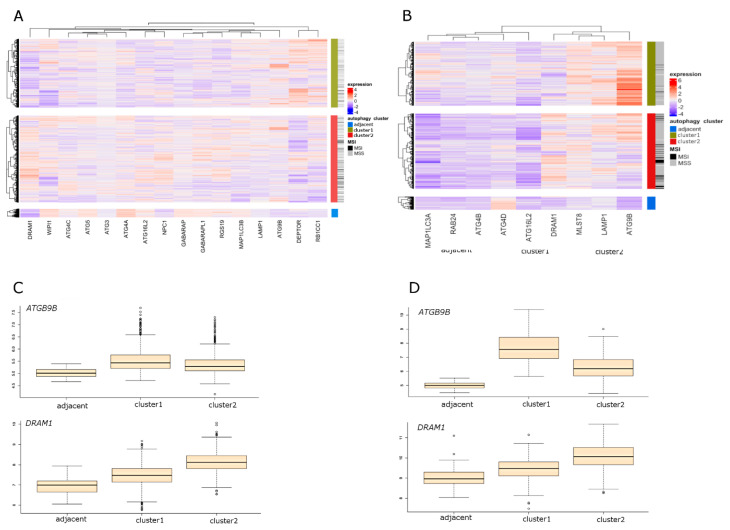
Results of unsupervised clustering and partial last squares discriminant analysis PLS (sPLS-DA) of colorectal cancer (CRC) datasets (RNA-seq and array) using selected autophagy-related genes. Samples were separated based on unsupervised clustering, and visualized genes were selected according to sPLS-DA [(**A**)—CRC array; (**B**)—CRC RNA-seq]. The corresponding box plots illustrate the expression distribution of the most highly ranked genes in terms of separation of clusters. [(**C**)—CRC array; (**D**)—CRC RNA-seq]. MSI: microsatellite instability; MSS: microsatellite stability.

**Figure 2 ijms-21-07101-f002:**
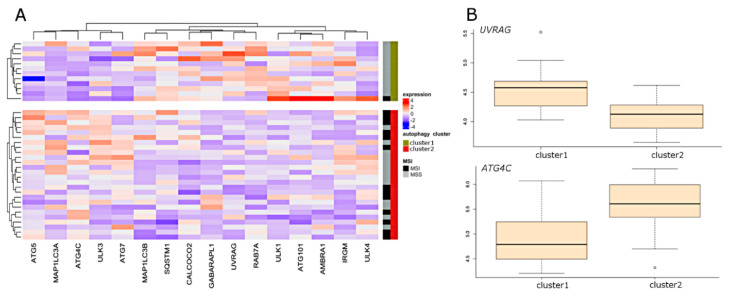
Results of unsupervised clustering and sPLS-DA in cell line dataset using selected autophagy-related genes. (**A**) Samples were separated based on unsupervised clustering and visualized genes were selected according to sPLS-DA. (**B**) The corresponding box plots illustrate the most highly ranked genes in terms of separation of clusters. MSI: microsatellite instability; MSS: microsatellite stability.

**Figure 3 ijms-21-07101-f003:**
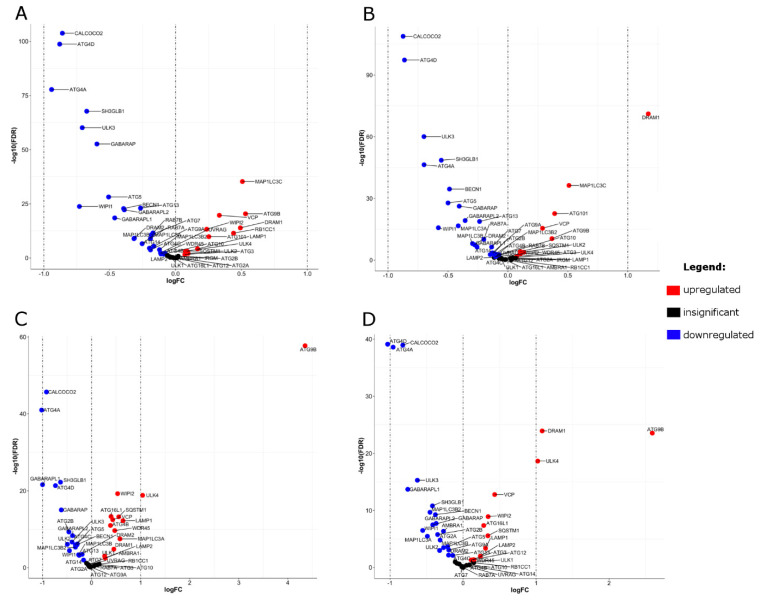
Differential expression (DE) analysis between autophagy-based clusters and normal adjacent tissues. (**A**,**B**) depict volcano plots based on DE analysis in the ‘limma’ package in CRC array dataset (**A**—cluster 1 vs. normal; **B**—cluster 2 vs. normal). (**C**,**D**) depicts volcano plots based on DE analysis in the limma package (RNA-seq datasets). (**C**—cluster 1 vs. normal; **D**—cluster 2 vs. normal). The vertical lines correspond to 1.0-fold up and down, respectively. The blue dot in the plot represents the gene with a False Discovery Rate (FDR) ≤ 0.05 and logFC < −1, whereas the red dot represents the gene with a FDR ≤ 0.05 and logFC > 1. Black dots indicate insignificant changes in gene expression. Data used to build volcano plots are included in Appendix A.

**Table 1 ijms-21-07101-t001:** Molecular characteristics of autophagy clusters (The Cancer Genome Atlas (TCGA) RNA-seq data).

Clinicopathological Characteristics	[ALL]	Cluster 1	Cluster 2	adj. *p*-Value *	*N* **
	*N* = 502	*N* = 232	*N* = 270		
MSI:				<0.001	497
MSI	71 (14.3%)	11 (4.8%)	60 (22.2%)		
MSS	426 (85.7%)	216 (95.2%)	210 (77.8%)		
CMScaller:				<0.001	457
CMS1	83 (18.2%)	18 (9.0%)	65 (25.4%)		
CMS2	141 (30.9%)	104 (51.7%)	37 (14.5%)		
CMS3	82 (17.9%)	21 (10.4%)	61 (23.8%)		
CMS4	151 (33.0%)	58 (28.9%)	93 (36.3%)		
age	68.0 [58.0; 76.0]	68.0 [59.0; 75.0]	68.0 [57.0; 77.0]	0.828	502
location:				<0.001	487
distal	273 (56.1%)	149 (66.2%)	124 (47.3%)		
proximal	214 (43.9%)	76 (33.8%)	138 (52.7%)		
stage:				0.003	486
I	84 (17.3%)	27 (12.1%)	57 (21.8%)		
II	189 (38.9%)	84 (37.5%)	105 (40.1%)		
III	138 (28.4%)	67 (29.9%)	71 (27.1%)		
IV	75 (15.4%)	46 (20.5%)	29 (11.1%)		
gender:				0.941	502
female	227 (45.2%)	104 (44.8%)	123 (45.6%)		
male	275 (54.8%)	128 (55.2%)	147 (54.4%)		
Single nucleotide variants	54.0 [41.0; 80.0]	51.0 [37.0; 66.0]	60.0 [41.0; 303.5]	<0.001	389
Non-silent mutation rate	2.7 [2.1; 3.8]	2.6 [1.9; 3.3]	2.9 [2.2; 5.0]	<0.001	382
Aneuploidy score	10.0 [4.0; 16.0]	11.0 [7.0; 16.0]	8.0 [2.0; 16.0]	<0.001	469
APCmut:				<0.001	422
0	96 (22.7%)	24 (13.0%)	72 (30.4%)		
1	326 (77.3%)	161 (87.0%)	165 (69.6%)		
TP53mut:				<0.001	422
0	172 (40.8%)	45 (24.3%)	127 (53.6%)		
1	250 (59.2%)	140 (75.7%)	110 (46.4%)		
KRASmut:				0.837	422
0	252 (59.7%)	112 (60.5%)	140 (59.1%)		
1	170 (40.3%)	73 (39.5%)	97 (40.9%)		
BRAFmut:				<0.001	422
0	365 (86.5%)	175 (94.6%)	190 (80.2%)		
1	57 (13.5%)	10 (5.4%)	47 (19.8%)		

* *p*-values adjusted using overall comparisons employing the Benjamini–Hochberg. ** samples with available data. MSI: microsatellite instability; MSS: microsatellite stability.

**Table 2 ijms-21-07101-t002:** Data of colorectal cancer patients (Affymetrix array data).

Clinicopathological Characteristics	[ALL]	Cluster 1	Cluster 2	adj. *p*-Value *	*N* **
	*N* = 1229	*N* = 505	*N* = 724		
age	69.0 [60.0; 77.0]	68.0 [59.0; 75.0]	70.0 [60.0; 78.0]	0.022	566
gender:				0.302	703
female	304 (43.2%)	135 (41.0%)	169 (45.2%)		
male	399 (56.8%)	194 (59.0%)	205 (54.8%)		
stage:				0.081	603
I	70 (11.6%)	31 (10.9%)	39 (12.3%)		
II	265 (43.9%)	119 (41.8%)	146 (45.9%)		
III	189 (31.3%)	87 (30.5%)	102 (32.1%)		
IV	79 (13.1%)	48 (16.8%)	31 (9.7%)		
MSI_native:				<0.001	685
MSI	162 (23.6%)	32 (10.4%)	130 (34.6%)		
MSS	523 (76.4%)	277 (89.6%)	246 (65.4%)		
CMScaller:				<0.001	1229
CMS1	218 (17.7%)	37 (7.3%)	181 (25.0%)		
CMS2	373 (30.3%)	254 (50.3%)	119 (16.4%)		
CMS3	215 (17.5%)	87 (17.2%)	128 (17.7%)		
CMS4	423 (34.4%)	127 (25.1%)	296 (40.9%)		

* *p*-values adjusted using overall comparisons employing the Benjamini–Hochberg. ** Samples with available data. MSI: microsatellite instability.

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
