# Peer review of "Transcriptomic Profiling for the Autophagy Pathway in Colorectal Cancer"

_ijms, 2020, doi:10.3390/ijms21197101_

Round 1
Reviewer 1 Report
The current study presents an interesting approach that may lead to identification of autophagy-associated genes as novel targets for precision medicine in CRC in the future. The work presents a promising concept in which modern machine learning tools were applied to determine the significance of expression of a panel of autophagy-associated genes in CRC. However, the following issues must be addressed in order to improve its presentation.
1. Please describe abbreviations full name at the first time they appear in text, e.g. sPLS-DA: Line 107.
2. Introduction: The introduction contains very little specific, and generic information, on why autophagy per se is a potential pathway for gene expression analysis in CRC.
3. Results: The presentation of the results should also be modified. In many places, one has to keep going back and forth between the Figures, e.g. the results section starts with description of Supp Table2, Then 2 panels of Fi. g1 and 1 panel of Fig 2 are described. Please either change the arrangement of Figures or discuss Figures as they appear in the paper.
4. Discussion: The discussion does not include any comparisons or mention of previous studies of autophagy-related gene expression analyses in CRC or other cancers. The discussion should include what are the strengths of this study compared to previous ones (e.g. Genome-Wide Identification of a Novel Autophagy-Related Signature for Colorectal Cancer, Huang et al.).
While the authors state that this study identifies autophagy-related genes as potential targets in CRC, no prognostic value was observed in terms of overall survival (OS) in CRC, please discuss this aspect.
Author Response
Wroclaw 16thSeptember, 2020
International Journal of Molecular Sciences
Assistant Editor
Diana Marie
Dear Editor,
Thank you for your efforts in processing our paper.
We greatly appreciate reviewers’ thoughtful comments, which we believe have helped us to improve the paper. Below we provide a point-by-point response to these comments. All changes in the manuscript have been marked in blue font.
Yours sincerely,
Justyna Gil
Reviewer 1
Comments and Suggestions for Authors
The current study presents an interesting approach that may lead to identification of autophagy-associated genes as novel targets for precision medicine in CRC in the future. The work presents a promising concept in which modern machine learning tools were applied to determine the significance of expression of a panel of autophagy-associated genes in CRC. However, the following issues must be addressed in order to improve its presentation.
- Please describe abbreviations full name at the first time they appear in text, g.sPLS-DA: Line 107.
Thank you. We corrected it.
- Introduction: The introduction contains very little specific, and generic information, on why autophagy per se is a potential pathway for gene expression analysis in CRC.
Thank you. We added information about autophagy and its contribution to CRC development and its importance for anticancer treatment.
- Results: The presentation of the results should also be modified. In many places, one has to keep going back and forth between the Figures, g.the results section starts with description of Supp Table2, Then 2 panels of Fi. g1 and 1 panel of Fig 2 are described. Please either change the arrangement of Figures or discuss Figures as they appear in the paper.
Thank you. We agree with the reviewer. The previous version was hard to follow. Corrected version is modified in numerous sections to increase readability. In each Results subsection we gave 1-2 introductory sentences. Some sentences in these sections were also rearranged. Figure 1 was rearranged. Figures 1, 2, 3 descriptions were also corrected.
- Discussion: The discussion does not include any comparisons or mention of previous studies of autophagy-related gene expression analyses in CRC or other cancers. The discussion should include what are the strengths of this study compared to previous ones (g.Genome-Wide Identification of a Novel Autophagy-Related Signature for Colorectal Cancer, Huang et al.).
Thank you. We included the mentioned article in the discussion section. We highlighted the strengths of our study in comparison to previous ones.
While the authors state that this study identifies autophagy-related genes as potential targets in CRC, no prognostic value was observed in terms of overall survival (OS) in CRC, please discuss this aspect.
Thank you. In our study , we did not study influence of single autophagy genes on overall-survival (OS) of CRC patients. We examined whether two CRC clusters defined by expression of autophagy-related genes differ in terms of OS. We found no differences in OS between CRC clusters, however, it does not mean that given autophagy gene can not be potential target of anti-cancer therapy even if its expression has no significant association with OS. Consequently, in our manuscript we stated that “Genes differentiating these clusters are promising candidates for CRC targeting therapy worthy of further investigation”.
Reviewer 2 Report
In their manuscript entitled “Genome-Scale Profiling for the Autophagy Pathway in Colorectal Cancer”, Gil et al. reported their potentially interesting findings that transcriptomic alterations of autophagy genes may serve as biomarkers for colorectal cancer. However, the manuscript needs to be written with more clarity so as to be properly evaluated.
- Utilizing RNA-Seq and microarray datasets for CRC tumor and adjacent tissues as well as CRC cell lines is revealing. However, the Results were poorly described. The rationale for each analysis and the connections among the analyses are unclear. A flow chart will be helpful to clarify what datasets were analyzed through what steps by what methods to reach what conclusions.
- In 2.1., it is unclear what datasets were used for EGSEA analysis in the Results itself. Are there only six autophagy-related gene sets in EGSEA? Number of gene sets probably does not mean much as it also depends on other factors such as how distinct each gene set is. Could the reason for the relative low ranks of autophagy-related gene sets be because the number of unique autophagy genes are small?
- In 2.2., it is unclear how were samples separated into two clusters based on the expression of the autophagy genes. Why were the two clusters MSI-enriched versus MSI-depleted? How were MSI and MSS brought into picture? As this seems to be the most important analysis in the paper, it is necessary to explain briefly what were done rather than just saying using the COMMUNAL approach. The reader cannot get from the Results but can only guess that unsupervised clustering might be done after sPLS-DA. Were 46 selected autophagy genes used in this analysis besides the sPLS-DA analysis?
- Figures 1 and 2. Were they done by PLS-DA or sPLS-DA? The method section has only sPLS-DA. If it was PLS-DA, please spell out PLS-DA and describe the method briefly.
- Tables 1 and 2: It is unclear how clusters 1 and 2 were divided, and what p.overall and N* were. In another word, how was the analyses done?
- In 2.5., what are autophagy clusters and how were genes that best differentiate between autophagy clusters selected?
- In 2.6., how was cluster-specific deregulation defined? Abs(LogFC) for ATG9B and DRAM1 is not always >1, depending on which excel sheet in Supplementary Table 3. GABARAPL1, ATG4D, ATG4A appeared to have Abs(LogFC)>1 in some cases. BTW, is Log with base 2?
- Do your results support a tumor suppressing or promoting function of autophagy in CRC?
Minor points:
- Although the title says “genomic profiling”, the datasets were transcriptomic.
- Line 23: What is a ‘defect’?
- Please correct NumGenes that were incorrectly spelt by excel as month-date in the Supplementary Table 2.
- In 2.3. and 2.4., the comparisons in parentheses were confusing as the orders of clusters 1 and 2 kept changing.
- How were outliers defined by means of PCA during data pre-processing?
- Spell out ‘RMA’.
Author Response
Wroclaw 16thSeptember, 2020
International Journal of Molecular Sciences
Assistant Editor
Diana Marie
Dear Editor,
Thank you for your efforts in processing our paper.
We greatly appreciate reviewers’ thoughtful comments, which we believe have helped us to improve the paper. Below we provide a point-by-point response to these comments. All changes in the manuscript have been marked in blue font.
Yours sincerely,
Justyna Gil
Reviewer 2
Comments and Suggestions for Authors
In their manuscript entitled “Genome-Scale Profiling for the Autophagy Pathway in Colorectal Cancer”, Gil et al. reported their potentially interesting findings that transcriptomic alterations of autophagy genes may serve as biomarkers for colorectal cancer. However, the manuscript needs to be written with more clarity so as to be properly evaluated.
- Utilizing RNA-Seq and microarray datasets for CRC tumor and adjacent tissues as well as CRC cell lines is revealing. However, the Results were poorly described. The rationale for each analysis and the connections among the analyses are unclear. A flow chart will be helpful to clarify what datasets were analyzed through what steps by what methods to reach what conclusions.
Thank you. We agree. The previous version was quite unclear. We have modified the manuscript in numerous sections to increase readability. In each Results subsection we gave 1-2 introductory sentences. Some sentences in these sections were also rearranged. Figure 1 has been rearranged. Figures 1, 2, 3 descriptions have been also corrected.
- In 2.1., it is unclear what datasets were used for EGSEA analysis in the Results itself. Are there only six autophagy-related gene sets in EGSEA? Number of gene sets probably does not mean much as it also depends on other factors such as how distinct each gene set is. Could the reason for the relative low ranks of autophagy-related gene sets be because the number of unique autophagy genes are small?
Thank you. EGSEA was performed only on the RNAseq dataset (this has been stated in the corrected version) as this method has been optimized to be used with RNAseq data. Yes, the definition of gene sets we used [Human Reactome definitions, January 2020] there are 6 gene-sets tagged “autophagy”. We are aware that this number has little meaning for results. We use this information to show that all autophagy related sets were ranked low. We utilized 4 enrichment algorithms to avoid any statistical bias. In Supplementary Table 1 the reader will find a column “NumGenes” that shows the number of genes found and number of genes defined in given gene-set. Looking at these results we do not see the relation between number of genes gene-set and position in ranking. Also, autophagy related gene-sets are average in terms of gene content.
- In 2.2., it is unclear how were samples separated into two clusters based on the expression of the autophagy genes. Why were the two clusters MSI-enriched versus MSI-depleted? How were MSI and MSS brought into picture? As this seems to be the most important analysis in the paper, it is necessary to explain briefly what were done rather than just saying using the COMMUNAL approach. The reader cannot get from the Results but can only guess that unsupervised clustering might be done after sPLS-DA. Were 46 selected autophagy genes used in this analysis besides the sPLS-DA analysis?
Thank you. We agree. The previous version was unclear. The current version was corrected and rearranged to enhance readability. We also added Supplementary Table 4 (summarizing a wrorkflow) to this submission. In short, first we selected 46 autophagy related genes. Secondly we used COMMUNAL to find two autophagy clusters. Third we used sPLS-DA to find genes that best separate “autophagy clusters”. Additionally we used classical DE analysis (using whole transcriptomes) to illustrate differences/similarities in gene expression of 46 autophagy genes between autophagy clusters.
- Figures 1 and 2. Were they done by PLS-DA or sPLS-DA? The method section has only sPLS-DA. If it was PLS-DA, please spell out PLS-DA and describe the method briefly.
Thank you. You are right. sPLS-DA –we corrected it. Figure 1 was re-arranged. Figure 1 and 2 descriptions were corrected.
- Tables 1 and 2: It is unclear how clusters 1 and 2 were divided, and what p.overall and N* were. In another word, how was the analyses done?
Thank you. We described it in the previous version in M&M section. N* is explained in Table footnotes. But you are right with “p.overall” –we corrected it. It was replaced by adj. p-value – also explained in table footnotes.
- In 2.5., what are autophagy clusters and how were genes that best differentiate between autophagy clusters selected?
Thank you. Autophagy clusters are output of COMMUNAL algorithm for which (as input) expression of 46 autophagy genes were selected. Subsequently these 46 genes were applied to sPLS-DA to find those which define autophagy clusters most efficiently.
We are aware that the previous version was not clear about it but in the corrected version we tried to clarify all.
- In 2.6., how was cluster-specific deregulation defined? Abs(LogFC) for ATG9B and DRAM1 is not always >1, depending on which excel sheet in Supplementary Table 3. GABARAPL1, ATG4D, ATG4A appeared to have Abs(LogFC)>1 in some cases. BTW, is Log with base 2?
Thank you. We used limma package to calculate DE between contrasts: autophagy cluster 1 – normal adjacent and autophagy cluster 2 – normal adjacent. This has been stated at least twice in newest version of the manuscript. And yes logFC is log2 fold change (“logFC” is original naming from limma package)
Indeed expression of some gens above (abs) logFC=1. This has been corrected and suitable information has been provided in results section.
- Do your results support a tumor suppressing or promoting function of autophagy in CRC?
Thank you. Our goal was not to determine if autophagy play a suppressing or promoting function in CRC because of its context-dependent manner. Our study is descriptive and we are, unfortunately, a long way from this definition being made. We added this information in study limitations section.
Minor points:
- Although the title says “genomic profiling”, the datasets were transcriptomic.
Thank you. The tittle has been corrected to “transcriptomic”.
- Line 23: What is a ‘defect’?
Thank you. We meant “field defect” as, by definition, a pre-malignant tissue with cellular and/or molecular alterations in which new cancers are likely to arise (e.g. Paul Lochhead et al. 2015, Etiologic Field Effect: Reappraisal of the Field Effect Concept in Cancer Predisposition and Progression, Mod Pathol. 2015 Jan; 28(1): 14–29.). For readability we changed it into “field cancerization”.
- Please correct NumGenes that were incorrectly spelt by excel as month-date in the Supplementary Table 2.
Thank you, it is corrected.
- In 2.3. and 2.4., the comparisons in parentheses were confusing as the orders of clusters 1 and 2 kept changing.
Thank you, it is corrected.
- How were outliers defined by means of PCA during data pre-processing?
Thank you. We used automatic outlier removal based on PCDIST algorithm (it is PCA-based) published in Information Sciences 245 (2013) 4–20. We clarified this in the corrected version of the manuscript.
- Spell out ‘RMA’.
Thank you, it is corrected.